# Comparative Study of Leak Detection in PVC Water Pipes Using Ceramic, Polymer, and Surface Acoustic Wave Sensors

**DOI:** 10.3390/s23187717

**Published:** 2023-09-07

**Authors:** Najah Hamamed, Charfeddine Mechri, Taoufik Mhammedi, Nourdin Yaakoubi, Rachid El Guerjouma, Slim Bouaziz, Mohamed Haddar

**Affiliations:** 1Laboratoire d’Acoustique de l’Université du Mans (LAUM), UMR CNRS 6613, 72085 Le Mans, France; charfeddine.mechri@univ-lemans.fr (C.M.); taoufik.mhammedi@univ-lemans.fr (T.M.); rachid.elguerjouma@univ-lemans.fr (R.E.G.); 2Laboratoire de Mécanique, Modélisation et Production (LA2MP), Ecole Nationale d’Ingénieurs de Sfax, LR13ES25, Sfax 1173-3038, Tunisia; slim.bouaziz@enis.tn (S.B.); mohamed.haddar@enis.rnu.tn (M.H.)

**Keywords:** detection of leaks, plastic pipes, polymer sensor, ceramic sensor, SAW sensor

## Abstract

The detection and location of pipeline leakage can be deduced from the time arrival leak signals measured by acoustic sensors placed at the pipe. Ongoing research in this field is primarily focused on refining techniques for accurately estimating the time delays. This enhancement predominantly revolves around the application of advanced signal processing methods. Additionally, researchers are actively immersed in the utilization of machine learning approaches on vibro-acoustic data files, to determine the presence or absence of leaks. Less attention has been given to evaluating the sensitivity, performance, and overall effectiveness of these sensors in leak detection; although acoustic methods have been successfully used for leak detection in metallic pipes, they are less effective in plastic pipes due to the high attenuation of leak noise signals. The primary thrust of this research centers on identifying sensors that not only possess sensitivity but also exhibit high efficiency. To accomplish this goal, we conducted an exhaustive evaluation of the performance of three distinct categories of acoustic sensors employed for detecting water leaks in plastic pipes: specifically, lead zirconate titanate (PZT) sensors, polyvinylidene fluoride (PVDF) sensors, and surface acoustic wave (SAW) sensors. Our evaluation encompassed the performance of PVDF and SAW sensors in leak detection, comparing them to PZT sensors under a variety of conditions, including different leak sizes, flow rates, and distances from the leak. The results showed that all three sensors, when they were placed in the same position, were able to detect water leaks in plastic pipes with different sensitivities. For small leaks (1 mm, 2 mm), the PVDF sensor showed the greatest sensitivity (0.4 dB/L/h, 0.33 dB/L/h), followed by the SAW sensor (0.16 dB/L/h, 0.14 dB/L/h), and finally the PZT (0.13 dB/L/h, 0.12 dB/L/h). Similarly, for larger leaks (4 mm, 10 mm), the PVDF sensor continued to show superior sensitivity (0.2 dB/L/h, 0.17 dB/L/h), followed by the SAW sensor (0.13 dB/L/h, 0.11), and finally the PZT sensor (0.12 dB/L/h, 0.1 dB/L/h), outperforming the PZT sensor. This suggests that SAW and PVDF sensors, have the potential to serve as valuable, cost-effective alternatives to traditional commercial leak noise transducers. The outcomes of this comparative study involving three acoustic sensors hold the potential to advance the development of robust and dependable systems for the detection of water leaks in plastic pipelines.

## 1. Introduction

Water leakage in water supply systems is causing substantial financial losses on a global scale. A significant portion of these supply pipelines is situated underground, allowing leaks to persist unnoticed for extended periods. The economic impact of this issue is significant, with an estimated annual water loss of 126 billion cubic meters conservatively valued at $39 billion in 2019 [1]. Access to clean and safe drinking water is recognized as a fundamental human right. Nevertheless, even today, roughly half of the global population experiences water scarcity at least once a month each year. Projections indicate that this number could rise to approximately 4.8 to 5.7 billion people by 2050 [2]. Presently, the European Union faces an average water leakage rate of 24%, while the United States experiences a 12% rate. This substantial water loss has a ripple effect, impacting both the economy and public health. According to the International Water Loss Association (IWA) estimates, water losses from drinking water supply networks worldwide amount to 346 billion liters every day. Reducing water loss by at least 30% provides sufficient savings to supply treated water to 800 million people [3]. Undoubtedly, improving the efficiency of water distribution systems for effective water transport carries substantial benefits for society, the environment, and the economy. As a result, there is a growing demand for the development of cost-effective equipment dedicated to precisely identifying pipeline leaks.

Leak localization involves determining the suspected leak’s position before excavation and repair work. This process typically occurs alongside leakage management, which encompasses monitoring and control activities. Over time, various leak detection techniques have been developed, including acoustic methods, chemical tracers, infrared thermography, ground-penetrating radar, in-line leak detectors and more recently, optical fibers [4,5,6,7,8]. Comprehensive discussions regarding these technologies, including an analysis of their strengths and weaknesses, are available in references [9]. However, it’s worth noting that non-acoustic methods have certain limitations, including higher costs, complexity, and longer time requirements, which may potentially restrict their practical use in leak detection surveys. Consequently, acoustic methods continue to be the preferred choice for detecting and locating leaks in water distribution networks [10,11,12,13,14,15]. These methods are often grouped into invasive or non-invasive categories, primarily based on how the sensor is attached to the pipe. This classification is based on whether the sensors require insertion into the pipe, thereby falling into the invasive category. Invasive techniques involve deploying devices like hydrophones within pipes to capture audio data. However, our study primarily focuses on non-invasive and passive methods, exemplified by Acoustic Emission (AE). AE stands apart from measurement techniques that actively stimulate a system; instead, it relies on collecting and analyzing ambient noise generated by the structure under examination. AE serves as a Non-Destructive Testing method, with its core principle centered on detecting sound or acoustic waves resulting from the sudden release of stress or strain within a material or structure. These emissions typically arise when there’s a change in the internal integrity or condition of the material, such as the initiation or propagation of cracks, fractures, or other defects. The process of acoustic emission testing entails continuous monitoring and analysis of these acoustic signals to identify and precisely locate flaws or irregularities in materials or structures. Acoustic Emission often involves deploying multiple acoustic sensors, making it a widely utilized technique across various industries, including manufacturing, aerospace, civil engineering, and materials science. It serves as a valuable tool for assessing the structural integrity and safety of components and constructions.

Current research in this field is primarily centered on the refinement of time delay estimation techniques, with a notable emphasis on advanced signal processing methods [16,17,18,19,20]. Researchers are also increasingly working on Machine Learning (ML)-based algorithms, to classify leak and non-leak sound [21]. Additionally, investigations are underway to explore different communication technologies, incorporating Internet of Things (IoT) systems into the research [11,13], to expedite leak response times and minimize water loss. However, it is crucial to underscore that there has been comparably limited focus on assessing the sensitivity, performance, and overall efficacy of sensors in leak detection. This becomes even more significant given that while acoustic methods have proven effective for detecting leaks in metal pipes, their performance is less pronounced in plastic pipes. This is primarily attributed to the substantial attenuation of leak sound signals in these pipelines [22,23,24,25,26,27].

The primary objective of our research is to identify AE sensors that not only demonstrate sensitivity, but also exhibit high efficiency. To achieve this goal, we conducted a comprehensive assessment of the performance of three distinct categories of acoustic sensors used in a passive way for detecting water leaks in plastic pipes: lead zirconate titanate (PZT) sensors, polyvinylidene fluoride (PVDF) sensors, and surface acoustic wave (SAW) sensors. Our evaluation encompassed the performance of PVDF and SAW sensors in leak detection, comparing them to PZT sensors under various conditions, including different leak sizes, flow rates, and distances from the leak.

PZT sensors and piezo-composites, based on Lead Zirconate Titanate material, are frequently employed in Acoustic Emission (AE) applications due to their notable characteristics, including a high piezoelectric coupling constant and a high mechanical quality factor [28]. In our study, we have chosen the PZT sensor because of its recognized performance in leak detection applications, as confirmed by the available literature. Its high sensitivity to acoustic signals and robustness in identifying leaks have made it a reference for such investigations [10,11]. Because of these well-documented advantages, we made the deliberate choice to incorporate the PZT sensor as a dependable reference point in our comparative study alongside PVDF and SAW sensors.

Polyvinylidene Fluoride (PVDF) is a piezoelectric polymer material renowned for its remarkable properties. These properties include flexibility, high sensitivity, a wide frequency response, low inertia, and cost-effectiveness [29]. These inherent advantages render PVDF transducers exceptionally versatile, finding application in various fields such as ultrasonic imaging, pressure sensing, accelerometry, and biomedical and underwater acoustics, among others. PVDF’s flexibility is a standout feature, allowing it to seamlessly adapt to diverse pipe configurations and shapes. It excels in establishing conformal contact with the intricate surfaces of plastic pipes [30,31,32]. This adaptability significantly enhances its potential for detecting leaks in various installation scenarios, making it a valuable addition to our study. This is especially beneficial when dealing with pipes that deviate from standard shapes and dimensions.

Surface Acoustic Wave (SAW) sensors are intricate devices designed to detect structural changes through surface acoustic waves [33,34,35]. These sensors consist of two key components: an interdigital transducer (IDT) and a piezoelectric substrate. When an electrical signal is applied, the piezoelectric substrate generates acoustic waves, and conversely, when functioning passively, the SAW can translate acoustic wave-induced changes in the structure. SAW sensors excel in detecting variations in wave propagation within a structure, brought about by alterations in temperature, pressure, humidity, or the presence of specific chemical ions. Their reputation is built on attributes such as remarkable sensitivity, rapid response times, and the capacity to measure a diverse array of physical properties [36]. Our preference for flexible SAW sensors in this leak detection study stems from several compelling rationales. Foremost, our choice is informed by the promising outcomes observed in PVDF film experiments. We recognized PVDF’s sensing potential and aimed to harness it by crafting a specialized SAW sensor. Unlike rigid sensors, flexible SAW sensors possess the ability to conform to intricate surfaces and irregular geometries. This adaptability facilitates integration into various systems, enhancing coverage and proximity to potential leakage sources. Consequently, this augments sensitivity and detection accuracy. This characteristic may allow precise measurements and the detection of faint acoustic emissions associated with leaks. Moreover, by introducing specific coatings or functionalizing the sensor surface, selectivity can be fine-tuned to target particular leak-inducing substances or gases [37,38]. Another remarkable feature of SAW sensors is their dual role as filtering devices. The inherent capacity to discriminate between various frequencies enables them to effectively filter out extraneous noise and interference. This ensures that only pertinent leak signals are detected. This integrated filtering capability minimizes false positives, bolstering the reliability and precision of leak detection, especially in intricate and noisy environments. Finally, it is worth highlighting that flexible passive Surface Acoustic Wave (SAW) sensors provide real-time monitoring capabilities. When these sensors are integrated wirelessly, they have garnered significant interest in various IoT-enabled applications. This wireless functionality enables the swift detection of leaks, potentially streamlining the implementation of timely intervention and mitigation measures. Consequently, this significantly bolsters the overall effectiveness of leak detection processes.

In this research, we conducted an assessment of the effectiveness of PVDF, PZT, and SAW sensors in detecting leaks within plastic pipes. While previous studies have examined the utility of PVDF and PZT sensors in this context, there exists a notable gap in the open literature regarding the specific evaluation of SAW sensors and their comparative performance with other sensor types for leak detection in plastic pipelines. Consequently, our study endeavors to bridge this knowledge gap and offer valuable insights into the application of acoustic sensors for the detection of water leaks.

## 2. Materials and Methods

In this study, we initially introduce the experimental setup and procedure designed to assess the sensitivity of various sensors on a plastic (PVC) pipe with a deliberately created leak area. To achieve this objective, we established the following steps:

Data Acquisition: We continuously recorded acoustic data using different types of sensors, employing a passive listening method. This approach allowed for uninterrupted monitoring of the specific area under investigation, offering a non-intrusive and highly informative means of data collection.

Pre-processing: This phase played a critical role in enhancing the quality of the collected data. Background noise, which had the potential to disrupt the signals, was meticulously removed. Furthermore, advanced signal processing techniques like bandpass filtering were applied during this stage. These techniques, implemented as a first-order analog filter, helped retain the essential elements of the signal while eliminating unwanted interference.

Spectral Processing: An analysis of the active frequency ranges within the acoustic signal was conducted, facilitating an in-depth exploration of spectral characteristics. This analysis phase allowed us to uncover distinctive signatures associated with the presence of the leak.

### 2.1. Sensors

To capture acoustic fields resulting from water leakage, various piezoelectric sensors were employed. One of these sensors was a flexible PVDF sensor, which took the form of a flexible film measuring 0.052 × 72 × 16 mm^3^, as depicted in Figure 1a. This sensor is constructed from piezoelectric film, possessing a broad bandwidth spanning from millihertz to gigahertz, characterized by high sensitivity, high dielectric strength, low acoustic impedance, and excellent stability. In this study, honey served as a coupling agent to adhere the PVDF sensor to the pipe’s surface. Honey, being a viscous liquid, facilitated both secure attachment of the sensor and close contact with the measuring surface. Additionally, a PZT sensor was employed, featuring a piezoelectric ceramic disk with a diameter of 20 mm and a thickness of 1 mm, as shown in Figure 1b. This sensor exhibits a flat response for radial acoustic modes in low frequencies (below 10 KHz). To affix the PZT sensor, Phenyl Salicylate (Salol) was used as the coupling agent. Salol is a crystalline organic compound that melts at a temperature of 43–45 °C, making it easy to apply in liquid form. As a coupling agent, Salol was spread on the tube’s surface, and the PZT sensor was pressed onto the molten Salol. Two clamps were used to secure this connection, preventing the formation of air gaps. Upon cooling, the Salol solidified, establishing a robust bond between the sensor and the tube’s surface. To ensure consistent coupling and maximize acoustic wave energy transmission into the water and pipe, pencil-lead break tests were conducted to assess the sensitivity of each sensor. Furthermore, in addition to these two sensors, a third type of sensor, the SAW sensor (Figure 1c), was used in our study.

As is well-established, a Surface Acoustic Wave (SAW) device comprises a piezoelectric layer upon which an interdigitated metallic comb-like structure, known as the Interdigitated Transducer (IDT), is meticulously fashioned. This structured arrangement of the IDT determines the wavelength of the acoustic wave generated on the surface of the piezoelectric substrate. By applying alternating electrical potentials to these two comb-like structures, they induce the formation of crests and troughs on the piezoelectric layer, resulting in the creation of a Surface Acoustic Wave (SAW). Notably, the device selectively permits only a limited range of frequencies to propagate through it, a phenomenon that is governed by the acoustic velocity within the piezoelectric material. This velocity can be calculated using the formula:$$f=\frac{V}{\lambda }$$
where ‘f’ corresponds to the resonance frequency, ‘*V*’ represents the acoustic velocity within the material, and ‘*λ*’ signifies the wavelength of the Surface Acoustic Wave (SAW). In our specific project, we undertook the fabrication of a flexible SAW device, utilizing a PVDF substrate, within the microtechnology facility at Le Mans University’s Laboratory of Acoustics (LAUM). This particular sensor, featuring a slender finger width (a) measuring 125 µm, a wavelength of 500 µm, and an acoustic velocity of 2500 m/s, exhibits a great sensitivity and offers a wide bandwidth, as clearly illustrated in Figure 1c. As a result, our sensor achieves a resonant frequency of 5 MHz. Furthermore, the categorization of Surface Acoustic Waves (SAWs) into distinct modes, such as Rayleigh waves, Lamb waves, Love waves, or Leaky waves, hinges on the properties of the piezoelectric substrate, particularly its thickness, denoted as ‘e’ (where e < *λ*). Considering our specific setup, where the substrate’s thickness conforms to the criterion (e < *λ*), we can confidently conclude that the acoustic wave modes detected by our SAW transducer predominantly correspond to Lamb modes.

### 2.2. Data Acquisition

Data acquisition was carried out through a Python program remotely controlling a Piscocope 4262 acquisition card. The card was configured with a sampling frequency of 100 KHz and a dynamic range of 16 bits. The acquisition process was executed in streaming mode, spanning a duration of 21 s.

### 2.3. Experimental Setup

In Figure 2, we have an illustration of the experimental arrangement. This depiction showcases a plastic polyvinyl chloride (PVC) pipe with an outer diameter of 32 mm and a thickness measuring 3 mm, which is connected to the water supply network. Along the surface of the pipe, sensors were strategically positioned on both sides of the leak. We conducted experiments at three distinct flow rates, spanning from low to high. In order to establish the voltage signature of a pipe in good condition, we recorded the output from PVDF sensors for five minutes at each flow rate. To ensure the reliability and reduce the impact of occasional outliers, these tests were repeated three times, resulting in robust statistical data. In order to simulate various leak scenarios, we drilled four holes with different dimensions and shapes in four separate PVC pipe samples. For each unique leak and sensor condition, we carried out tests, resulting in a total of 180 tests. This comprehensive set of tests encompassed five sensor conditions, four leak sizes, three flow rates, and each individual test was replicated three times. To mitigate the challenges posed by high attenuation and acoustic dissipation, we made a deliberate choice to employ relatively short tubes, each measuring 2 m in length. This decision was guided by the intention to minimize the potential impact of acoustic dissipation, as discussed in reference [37]. We strategically positioned sensors at various points along these tubes. Our objective was to explore how the distance factor influences signal attenuation, primarily because of substantial acoustic dissipation into the flexible substrate, while also assessing the sensors’ sensitivity to detect leaks. Table 1 provides a comprehensive listing of the various sensors utilized for the detection of acoustic radiation, complete with their respective acronyms.

In our research, we conducted experiments using a diverse set of samples that incorporated holes of varying sizes and shapes. We achieved this diversity by employing drill bits with different dimensions. We intentionally introduced artificial leaks of differing dimensions and shapes into the pipe to assess how these factors influenced both the amplitude level and the frequency ranges of the acoustic signals. For clarity, you can find a comprehensive breakdown of these sample tube tests in Table 2, and a visual representation is provided in Figure 3. These tables and figures assign unique numerical identifiers, ranging from 1 to 4, to denote the specific conditions of each sample tube test. As an example, when you see “sample S1”, it signifies a leak with a diameter of 1 mm.

### 2.4. Signal Preprocessing

Our experimental setup records acoustic data over a duration of 21 s, which is then divided into 3-s sequences. To select the most reliable signal, free from noise and disturbances, we employed two key criteria: total energy and temporal flatness coefficient. These criteria were computed for each sequence. The sequence displaying the best temporal flatness with the lowest energy levels (within a 3 dB range from the minimum) was considered the ideal candidate to represent the acoustic signal corresponding to the specific leak level. Detecting leaks in flowing water pipes presents challenges, as the acoustic emissions from the flowing water can often overshadow those generated by actual leaks. To tackle this challenge, we employed the Fast Fourier Transform (FFT) method. This technique provided us with quantitative insights into the acoustic signatures of pipe leaks, even in the presence of acoustic radiations induced by water flow. To enhance the signal analysis, we applied Blackman windowing, a method that minimizes ripple effects in the frequency domain, making it particularly suitable for the treatment of signals with broad frequency bands.

## 3. Results

The objective of this study is to assess the sensitivity of various sensors for detecting leaks in plastic water pipes. This section presents the findings obtained for different samples, each featuring varying geometries and sizes of leakage sources. In our investigation, we considered three distinct levels of water leakage flow rates, as detailed in Table 3.

### 3.1. Circular Defect Samples S1 and S2

#### 3.1.1. Acoustic Signatures

Figure 4 and Figure 5 illustrate the sensor responses, displaying the relationship between acoustic intensities and frequencies, for the two respective samples, S1 and S2.

In this study, we employed three PVDF sensors, namely C1, C3, and C5. C1 and C3 were strategically positioned at a fixed distance of 14 cm from the leakage source, with one on each side. This placement allowed us to assess the sensor’s response both before and after the leak. By comparing their performance, we aimed to gauge the sensors’ directivity concerning defect detection, particularly in Structural Health Monitoring (SHM) applications. Figure 4 and Figure 5 consistently reveal that for holes with diameters of both 1 and 2 mm, the frequency band encompassing acoustic activity falls within the [80 Hz–20 kHz] range. However, when observing PVDF sensor C5, positioned at a greater distance of 38 cm from the leak, we notice a significant drop in high frequencies. This results in a narrower frequency range, specifically [80 Hz–2 kHz], as evident in Figure 4b and Figure 5b. Figure 4d and Figure 5d shed light on the acoustic intensity of the two tested samples, S1 and S2, as flow rates vary. The spectral analysis uncovers that the PZT sensor, characterized by line contact, does not perform favorably in detection. This limitation arises because the rigid PZT ceramic disc is attached using a reversible glue (Salol) and makes axial contact with the pipe, rendering it primarily sensitive to axial modes.

Conversely, the SAW sensor exhibits sensitivity at lower frequencies, commencing at 100 Hz. However, its responsiveness to acoustic activity significantly heightens within the [1 kHz–20 kHz] range, corresponding to the primary acoustic signatures captured by PVDF sensors.

#### 3.1.2. Sensors’ Sensitivities Comparison

Sensor sensitivities reflect their capacity to perceive and react to alterations in the observed parameter or quantity. Essentially, sensitivity quantifies how much a sensor’s output shifts in response to a specific alteration in the input. The outcomes of our measurements, obtained within the frequency spectrum spanning from 80 Hz to 20 KHz, have been concisely summarized and are presented in the subsequent figures. These figures depict the acoustic intensity in relation to the flow rate for each sensor. This presentation method facilitates a straightforward visual assessment and facilitates the comparison of sensor responses across varying flow rate intervals.

In Figure 6a, it is evident that the acoustic intensity, as measured by PVDF sensors, increases in direct proportion to the flow rate. This consistent trend holds true across various leakage scenarios, indicating that all sensors exhibit sensitivity to leak detection. However, it is important to note that the proportionality ratio decreases with distance due to attenuation effects. As the acoustic energy travels from sensor C3, positioned at 14 cm, to sensor C5, situated at 38 cm, the acoustic waves lose energy, resulting in less sensitive detection.

Table 4 provides us with the means to quantify this sensitivity decrease over distance, specifically from C3 to C5. The calculated decrease is approximately 0.55 [dB/(L/h)]/m with an error margin of around 9%. This measurement offers valuable insights for designing an industrial passive monitoring system for water leakage detection in residential settings. By strategically positioning sensors, we can maintain optimal performance. With the implementation of suitable filtering and amplification systems, it is feasible to ensure that the acoustic intensity remains within a 6 dB range over a distance of 12 m from the leakage source.

Figure 6b highlights a distinct characteristic: the relationship between acoustic intensity and leak flow rate for the PZT sensor does not exhibit the linearity observed in the PVDF sensor placed at the same location. Instead, the curve illustrating the trend of acoustic intensity concerning leak flow rate shows a nonlinear pattern for the PZT sensor. This nonlinearity can significantly contribute to the challenge of detecting minor leaks when relying on the PZT sensor. Moreover, when examining the data in Table 5, it becomes evident that the PZT sensor is notably less sensitive, with a sensitivity of 0.12 dB/L/h for S1 and 0.13 dB/L/h for S2, making it the least sensitive among the sensors. On the other hand, the SAW sensor displays a strong correlation between acoustic intensity and leakage flow rate. Notably, when the SAW sensor is positioned in the same location as a PVDF sensor, it exhibits a sensitivity of 0.16 dB/L/h for sample S1 and 0.14 dB/L/h for S2. In comparison, the SAW sensor is 50% less sensitive than PVDF films placed at the same location.

### 3.2. Large and Small Circumferential Slit Defects S3 and S4

#### 3.2.1. Acoustic Signatures

For these measurements, we designed slits with dimensions of 4 × 1 mm^2^ (S3) and 10 × 1 mm^2^ (S4), as seen in Figure 7 and Figure 8.

It is worth noting that the smaller slit (S3) has a leakage surface area four times larger than the circular defect in sample 1 and twice the size of S2. This accounts for the higher flow rates observed and, consequently, the increased levels of acoustic activity. Sensor C5, which previously struggled to detect leaks from S1 and S2, now effectively discerns the acoustic signature generated by these circumferential slits. The primary acoustic activity predominantly falls within the frequency range of [500 Hz–10 kHz]. Despite the heightened acoustic intensity, the PZT sensor encounters challenges in detecting the acoustic activity stemming from the slits. Figure 7d and Figure 8d offer a comparative view of acoustic intensity for both slit sizes across varying flow rates. Spectral analysis reveals that the recorded signals from the leaks primarily produce acoustic radiations within the frequency band of [1 kHz–10 kHz]. In Figure 7e and Figure 8e, we present the processed data derived from the leakages detected by the Surface Acoustic Wave (SAW) sensor. Much like our observations with circular defects, the SAW sensor consistently proves to be a highly reliable choice for identifying leakage signatures across different flow rates. The data analysis distinctly showcases patterns and characteristics associated with the identified leaks, further emphasizing the SAW sensor’s reliability and precision in detecting and localizing leaks within the tested system.

#### 3.2.2. Sensors Performance Comparison

In this section, we have conducted a comparative analysis of the performance of three distinct sensors: PVDF, PZT, and SAW, focusing on their effectiveness in detecting water leaks. This comparison is grounded in a comprehensive assessment of their capabilities and characteristics. The findings have been succinctly summarized in a comparison graph (Figure 9) and Table 6 and Table 7, which offer a lucid overview of each sensor’s performance. In Figure 9a, it is evident that the sensitivity of PVDF sensors to leaks increases proportionally with the flow rate. This trend remains consistent across various leak scenarios. However, the proportionality ratio decreases concerning distance due to attenuation. As acoustic energy travels from sensor C3, situated at 14 cm, to sensor C5, positioned at 38 cm, the acoustic waves lose energy, resulting in less sensitive detection. Referring to Table 5, we can quantitatively measure this sensitivity decrease over distance, from C3 to C5, yielding a decrease of approximately 0.33 [dB/(L/h)]/m, with an error margin of about 12%.

Figure 9b indicates that there is some improvement in linearity as the diameter of the leak increases. However, it is important to note that the relationship between acoustic intensity and leak flow rate for the PZT sensor does not display a strong linear pattern. The absence of strong linearity in this curve can be attributed to various factors, which we will delve into in Section 4. In contrast, the SAW sensor demonstrates a robust correlation between acoustic intensity and leakage flow rate, as depicted in Figure 9c. Consequently, we can conclude that the SAW sensor consistently maintains a 50% lower sensitivity compared to PVDF films in this configuration, and that SAW and PVDF remain more sensitive than PZT.

## 4. Discussions

Frequencies below 100 Hz typically remain unaffected by the leakage signals analyzed here. This is primarily because they are predominantly influenced by ambient noise and the longitudinal resonance frequency of the plastic pipe. However, frequencies ranging from 100 Hz to 10 KHz are of utmost significance across all types of leaks and sensors.

It is noteworthy to emphasize that the acoustic signals examined in this study primarily consist of relatively high frequencies. These findings align with numerous studies in the existing literature, which have concentrated on the detection of high-frequency acoustic signals. Such studies often utilize Acoustic Emission sensors that typically operate in the MHz range [39,40,41,42,43]. Based on the findings, it can be concluded that the size of the leak aperture is a significant factor in the dissipation of acoustic intensity. However, small leaks have a minimal impact on attenuating the acoustic intensity propagating beyond the leak aperture because small leaks generally have a smaller opening, which limits the flow of water and the turbulence generated. As a result, less energy is transferred from the leak, leading to the generation of a weaker acoustic signal.

The PZT sensor, compared with the PVDF sensor and the SAW sensor with a PVDF substrate, was less sensitive in detecting leaks when working at high frequencies. This is explained by the fact that PZT sensors generally have a limited frequency response range compared with PVDF and SAW sensors. High-frequency vibrations caused by leaks may lie outside the effective range of the PZT sensor, reducing sensitivity. Also, the PZT sensor’s compatibility issues with the shape of the tube can affect the accuracy of the measurements. If the sensor is not optimally positioned or in direct contact with the leak area, it may not capture the acoustic energy generated by the leak accurately, leading to deviations from linearity. Lastly, other environmental factors, such as background noise and variations in the flow dynamics within the pipe, can affect the PZT and other sensor’s response. This phenomenon can also affect the detection capabilities of a PVDF sensor located far from the leak. Therefore, Flexible SAW and PVDF sensors have a better response to detect leaks in PVC pipes. Consequently, it is clear that there are several parameters which can affect the detection of leaks in pipes:Effect of pipe’s material: The detection in plastic pipes is more difficult than metallic ones because of the high attenuation of acoustic propagation in plastic pipes;Effect of the distance between the crack and the sensor: It can be noticed that as the sensor moves away from the leak position, it detects less noise from the leak;Effect of pipe’s diameter: The diameter of the pipe can influence several factors related to leak detection, including the magnitude of the leak signal, the speed of leak propagation, and the sensitivity of the detection method;Effect of leak type: The size and shape of the leak affect the frequency range of leak signals;Effect of leak flow rate: The flow rate of the leak was increased by increasing the size of the leak. In addition, for all measurements, we remark that the flow rate and the amplitude of leak signals are proportional; the increase of the flow rate increases the signal’s amplitude.

The results obtained from the SAW sensor for leak detection are currently in the preliminary stage. Nevertheless, it is worth highlighting that the sensor displays promising sensitivity in leak detection, on par with that of PVDF film. An advantageous feature of the SAW sensor lies in its ability to act as a built-in filter, eliminating the need for an additional filter component. This capability stems from the inherent selectivity and sensitivity of SAW sensors to alterations in the sensing environment. In practical terms, this means that they can effectively differentiate between various types of signals and only detect those relevant to the leak detection system. Further improvements in the optimization process will contribute to enhancing the sensor’s performance, making it even more suitable for advanced leak detection applications.

## 5. Conclusions

In conclusion, our study delved into the realm of leak detection in plastic water pipes using acoustic intensity monitoring. It offered a comprehensive analysis of the sensitivity of various sensor types, namely PZT, PVDF, and SAW sensors, when applied to the same pipes and conditions. Furthermore, it is worth noting that, to the best of our knowledge based on the available literature, Surface Acoustic Wave (SAW) sensors have not been previously evaluated for their effectiveness in water leak detection. By elucidating the strengths and limitations of each sensor under specific circumstances, our research serves as a valuable resource for researchers and engineers engaged in the development of leak detection systems. Additionally, by revealing how sensors perform relative to different leak types, our study aids in making informed sensor selections tailored to specific application needs.

The results underscored the feasibility of acoustically identifying pipe leaks through sensor measurements. The frequency range of the acoustic signals emanating from leaks could vary based on leak dimensions and shapes. While all sensors utilized in this study demonstrated leak detection capabilities, the PZT sensor struggled with detecting faint leaks due to its limited frequency response range and incompatibility with the pipe’s shape. Conversely, the PVDF sensor and the flexible SAW sensor exhibited superior performance in detecting leaks, including weaker ones.

In light of these findings, it is evident that employing the SAW sensor for leak detection offers distinct advantages. Unlike its counterparts, the SAW sensor obviates the need for an additional filter during testing and exhibits sensitivity across a broad frequency spectrum. Nonetheless, further optimization is warranted to heighten its sensitivity.

However, it is crucial to acknowledge the limitations of our study. Firstly, the experimental setup was designed to simulate monitored leak detection conditions within a specific context. This may not capture all the intricacies of real-world pipe networks, such as buried pipes. Additionally, we did not account for other factors that could impact sensor performance, such as the composition of fluids transported within the pipes.

Our future endeavors will encompass experiments in more complex pipe networks with diverse configurations. We will also concentrate on refining and expanding the array of sensor types adaptable to intricate scenarios. Furthermore, we will prioritize enhancing the SAW sensor’s capabilities to achieve even greater sensitivity and precision in leak detection applications. Capitalizing on the potential of the SAW sensor and addressing its optimization holds the promise of significant advancements in the field of leak detection in plastic water pipes.

Continued investigations into sensor performance under diverse fluid conditions and within more intricate configurations hold promise for gaining valuable insights. Furthermore, advancing the development of data processing and analysis methods, harnessing the capabilities of high-performance sensors, has the potential to enhance our capacity for identifying, pinpointing, and addressing leaks more effectively.

## Figures and Tables

**Figure 1 sensors-23-07717-f001:**
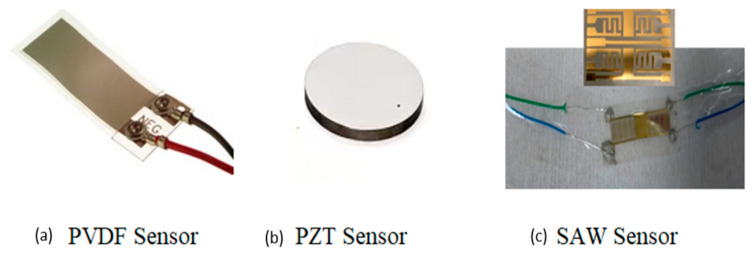
Sensors used in the study.

**Figure 2 sensors-23-07717-f002:**
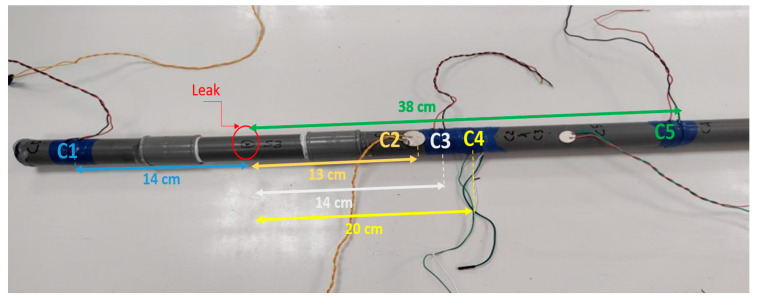
Water leak detection prototype: PVC pipe and sensors’ positions.

**Figure 3 sensors-23-07717-f003:**
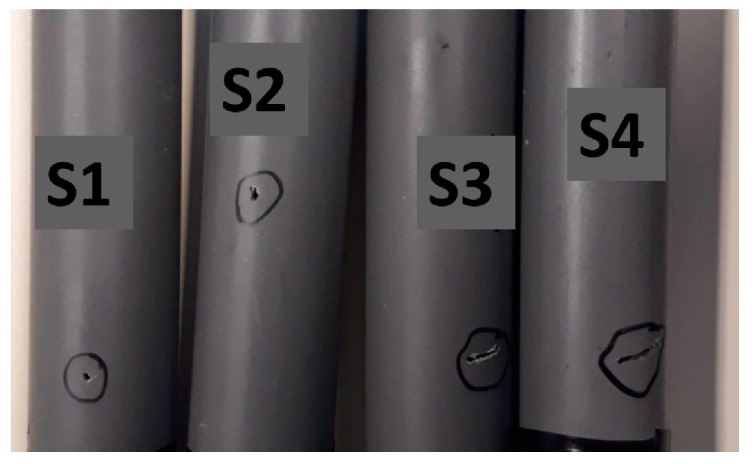
Samples for different leaks: S1 (1 mm), S2 (2 mm), S3 (4 × 1 mm^2^), S4 (10 × 1 mm^2^).

**Figure 4 sensors-23-07717-f004:**
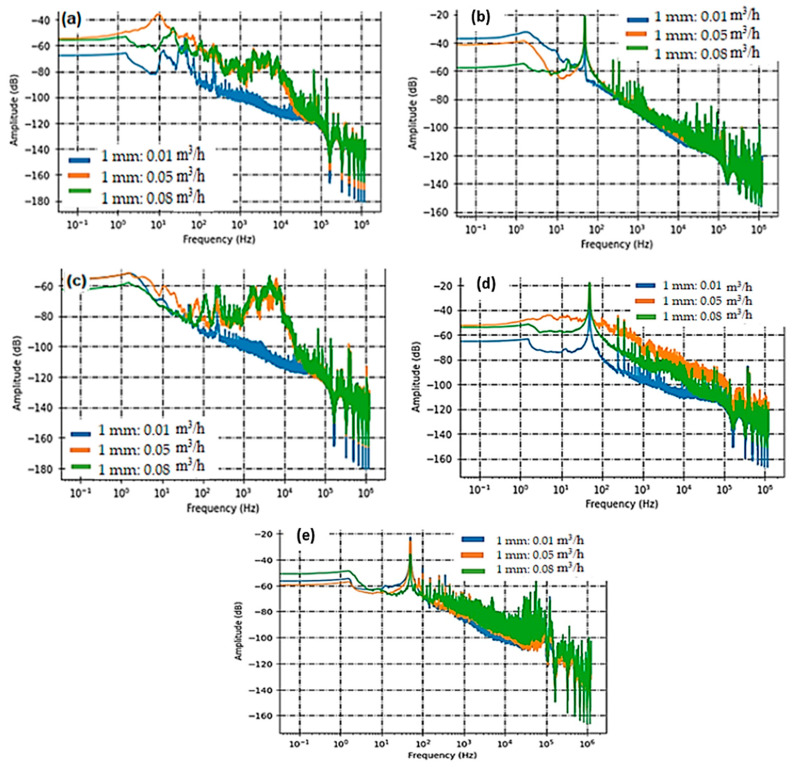
The acoustic intensity data, as captured by various sensors (PVDF Sensor C3 in (**a**), C5 in (**b**), C1 in (**c**), PZT Sensor C2 in (**d**), and SAW Sensor C4 in (**e**)), pertain to a leakage defect with a diameter of 1 mm.

**Figure 5 sensors-23-07717-f005:**
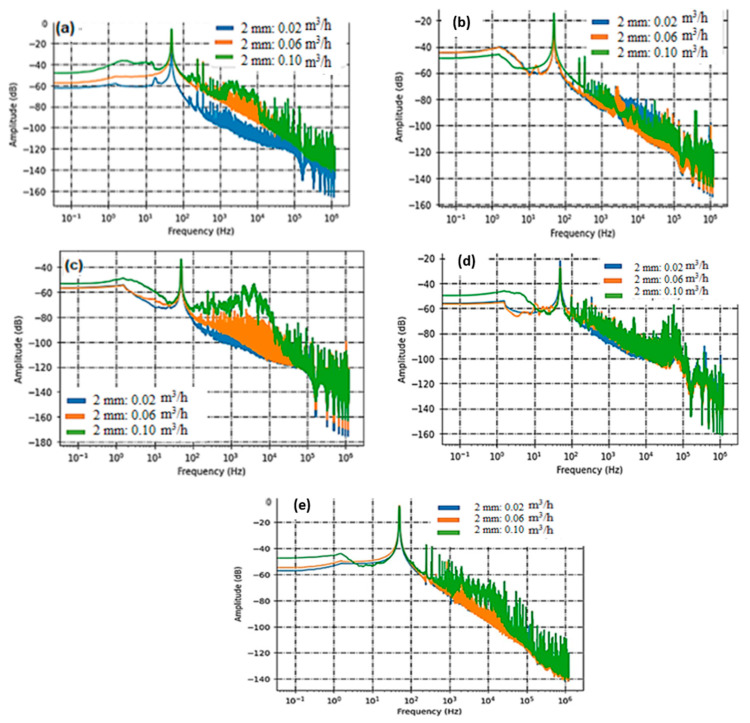
The acoustic intensity data, as captured by various sensors (PVDF Sensor C3 in (**a**), C5 in (**b**), C1 in (**c**), PZT Sensor C2 in (**d**), and SAW Sensor C4 in (**e**)), in the context of a leakage defect with a 2 mm diameter.

**Figure 6 sensors-23-07717-f006:**
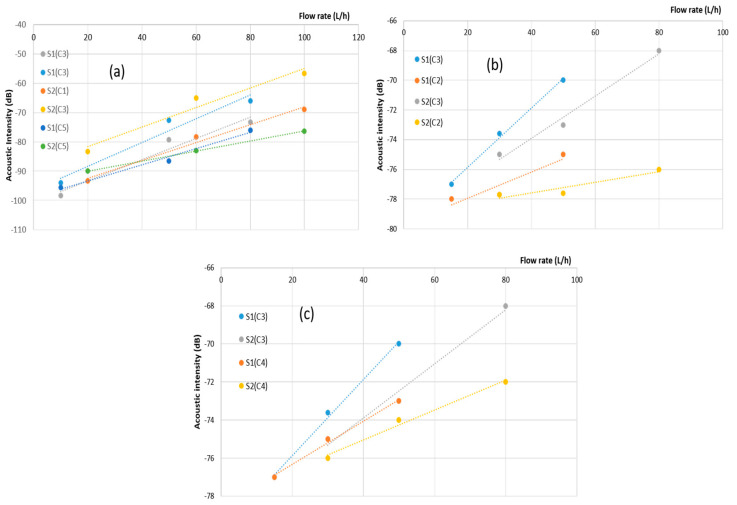
A condensed representation of the relationship between acoustic intensity and flow rate. It segregates the data into three subfigures: (**a**) for PVDF sensors (C1, C3, C5), (**b**) for the PZT sensor (C2), and (**c**) for the SAW sensor (C4).

**Figure 7 sensors-23-07717-f007:**
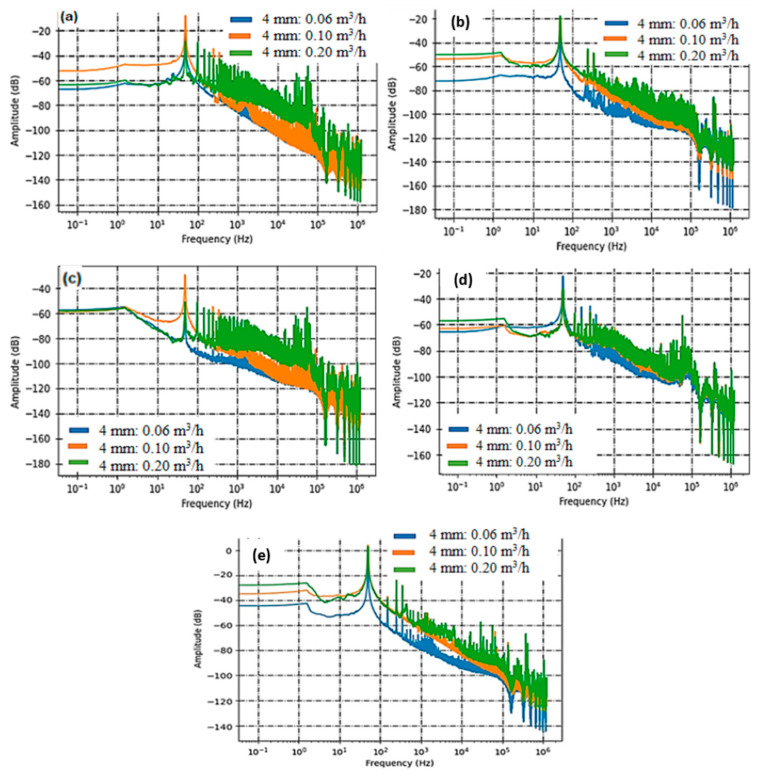
Acoustic intensity measured with PVDF Sensor C3 (**a**), C5 (**b**), C1 (**c**), PZT Sensor C2 (**d**), and SAW Sensor C4 (**e**) on rectangular circumferential leakage defect with widths of 4 mm.

**Figure 8 sensors-23-07717-f008:**
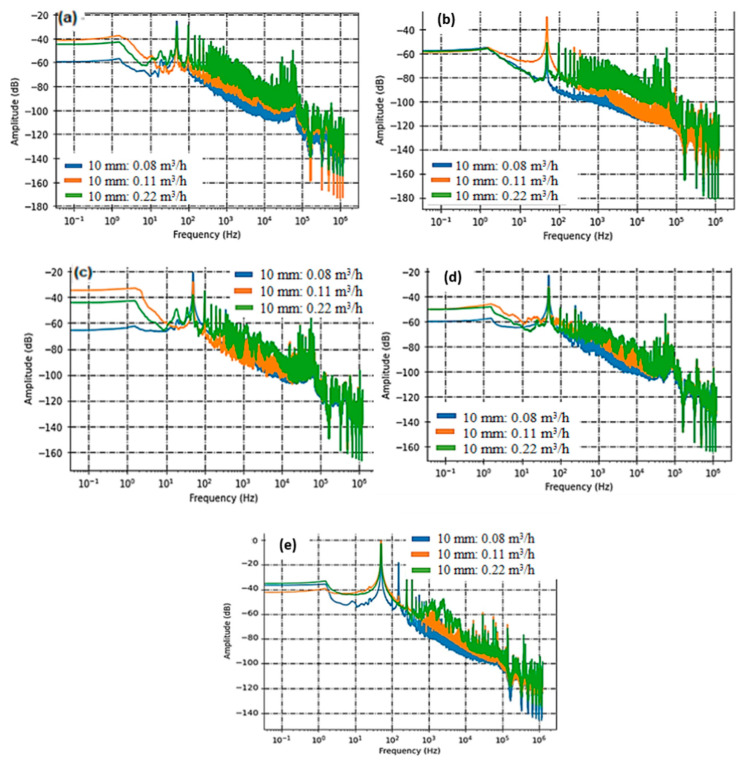
Acoustic intensity measured with PVDF Sensor C3 (**a**), C5 (**b**), C1 (**c**), PZT Sensor C2 (**d**), and SAW Sensor C4 (**e**) on rectangular circumferential leakage defect with widths of 10 mm.

**Figure 9 sensors-23-07717-f009:**
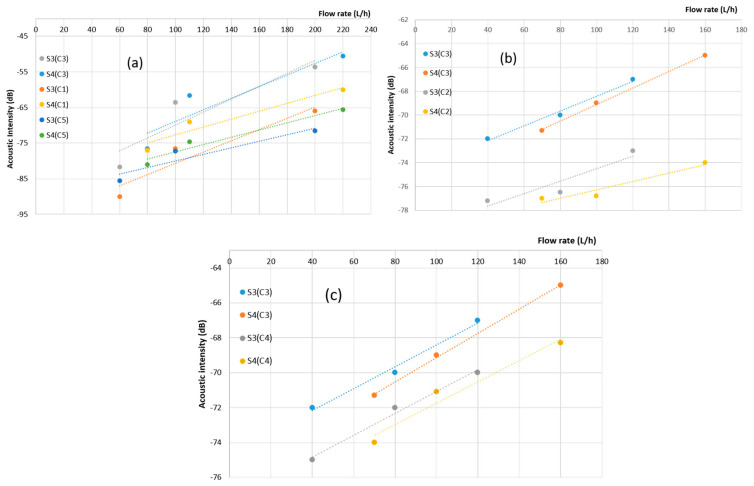
Summary of acoustic intensity versus flow rate for (**a**): PVDF (C1, C3, C5), (**b**): PZT (C2) sensors, and (**c**): SAW (C4).

**Table 1 sensors-23-07717-t001:** Sensors used in measurement.

Sensor	Type
C1	PVDF sensor
C2	PZT sensor
C3	PVDF sensor
C4	SAW sensor
C5	PVDF sensor

**Table 2 sensors-23-07717-t002:** Dimensions of leaks.

Samples	Shape and Dimensions of Leaks
S1	Diameter: 1 mm
S2	Diameter: 2 mm
S3	Small circumferential line: 4 × 1 mm^2^
S4	Large circumferential line: 10 × 1 mm^2^

**Table 3 sensors-23-07717-t003:** Flow rate (F.R) admitted by a leak.

Flow Rates	S1	S2	S3	S4
F.R. 1	0.01 $m^{3}/h$	0.02 $m^{3}/h$	0.06 $m^{3}/h$	0.08 $m^{3}/h$
F.R. 2	$0.05\;m^{3}/h$	$0.06\;m^{3}/h$	$0.1\;m^{3}/h$	$0.11\;m^{3}/h$
F.R. 3	0.08 $m^{3}/h$	0.1 $m^{3}/h$	0.2 $m^{3}/h$	0.22 $m^{3}/h$

**Table 4 sensors-23-07717-t004:** Comparison of PVDF sensors sensitivities positioned at different positions C1, C3, and C5 used for Circular Defect Samples S1 and S2.

	C3	C5	C1
Sensitivity on S1	0.4 dB/L/h	0.28 dB/L/h	0.35 dB/L/h
Sensitivity on S2	0.33 dB/L/h	0.18 dB/L/h	0.3 dB/L/h

**Table 5 sensors-23-07717-t005:** Comparison of PVDF, SAW, and PZT sensor sensitivities positioned at the same position, used for Circular Defect Samples S1 and S2.

	C3	C4	C2
Sensitivity on S1	0.4 dB/L/h	0.16 dB/L/h	0.12 dB/L/h
Sensitivity on S2	0.33 dB/L/h	0.14 dB/L/h	0.13 dB/L/h

**Table 6 sensors-23-07717-t006:** Comparison of PVDF sensors sensitivities positioned at different positions C1, C3, and C5 used for Circumferential Slit Defects S3 and S4.

	C3	C5	C1
Sensitivity on S3	0.2 dB/L/h	0.11 dB/L/h	0.17 dB/L/h
Sensitivity on S4	0.17 dB/L/h	0.1 dB/L/h	0.12 dB/L/h

**Table 7 sensors-23-07717-t007:** Comparison of PVDF, SAW, and PZT sensor sensitivities positioned at the same position, used for Circumferential Slit Defects S3 and S4.

	C3	C4	C2
Sensitivity on S3	0.2 dB/L/h	0.13 dB/L/h	0.12 dB/L/h
Sensitivity on S4	0.17 dB/L/h	0.11 dB/L/h	0.1 dB/L/h

## Data Availability

Not applicable.

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
