# Peer review of "Comparative Study of Leak Detection in PVC Water Pipes Using Ceramic, Polymer, and Surface Acoustic Wave Sensors"

_sensors, 2023, doi:10.3390/s23187717_

Round 1

Reviewer 1 Report

Minor English editing required.

Reviewer 2 Report

In this work, the authors proposed three sensors (PZT, PVDF and SAW) sensors) for water leak detection in plastic pipes and compared their performances through changes of flow rate, distance from the leak and the leak size and shape. This work is interesting and is useful for developing efficient and reliable water leak detection systems. Before considering this manuscript for publication, the following points should be addressed:

1.    The paragraphs in Introduction Section is fragmented, the authors should reorganize them. In the first section, the authors should introduce the harm of water leak in plastic pipes. In the second section, the authors can introduce the existed methods or technologies for water leak detection and their advantages and disadvantages. In the third section, the authors can introduce acoustic emission method for water leak detection and their advantages and why you chose these three sensors for water leak detection. The authors should modify them.

2.    In 2.1, Sensors Section, the authors introduce lots of contents about flexible SAW sensors, it would be better to put these in introduction section.

3.    For PVDF sensor, which acoustic wave mode do you use? How much is the resonant frequency?

4.    For PVDF sensor measurement, the authors chose honey as a coupling layer, but for PZT sensor, the authors chose Salol as a coupling layer. Why? What are the differences between them?

5.    As we all known, PZT sensor is rigid, so the contact between PZT sensor and the plastic pipe is linear. How can you couple more acoustic wave energy into the water and the pipe?

6.    For the SAW sensor, the authors used PVDF as the piezoelectric substrate. However, as far as I know, PVDF SAW sensor has a poor signal due to significant acoustic dissipation into flexible substrate. Therefore, it is hard to generate a good acoustic emission. Can you show the measured signal for PVDF SAW sensors and explain this? Which acoustic wave mode do you use and how much is the resonant frequency of the SAW device?

7.    In Experiment setup Section, the authors installed two sensors on both side of the leak. How the signal is measured? Do you use one sensor to transmit and another sensor to receive? If the acoustic wave is attenuated in liquid and plastic pipe, how can they be detected? Hence, the authors should introduce the acoustic wave modes you used and explain why?

8.    For the sensor performance comparison, these three sensors should put in the same position, such as in C3, otherwise, the comparison is meaningless. The authors should added these results.

9.    For sample S1, S2, S3 and S4, the authors should keep the flow rate of plastic pipe at the same, and studied the effect of leak size on sensor measurement results or keep the leak size at the same and study effect of flow rate on sensor measurement results. The results presented in Figures 4, 5, 6 and Figures 8, 9, 10 are chaotic.

10. For Figures 7b and 7c, the results is relatively few, the authors should supplement them.

11. There are some mistakes in the manuscript, such as figure 7.a in line 252, figure 7.c in line 268, etc.

12. The references are relatively old, the authors should update it.

The English is poor and need to improve.

Round 2

Reviewer 2 Report

none